The taxonomy of a new parvicursorine alvarezsauroid specimen IVPP V20341 (Dinosauria: Theropoda) from the Upper Cretaceous Wulansuhai Formation of Bayan Mandahu, Inner Mongolia, China

Pittman Michael 1 mpittman@hku.hk
Xu Xing 2
Stiegler Josef B. 3
1 Vertebrate Palaeontology Laboratory, Life and Planetary Evolution Research Group, Department of Earth Sciences, The University of Hong Kong , Pokfulam , Hong Kong
2 Key Laboratory of Vertebrate Evolution and Human Origins, Institute of Vertebrate Paleontology & Paleoanthropology, Chinese Academy of Sciences , Beijing , China
3 Department of Biological Sciences, George Washington University , Washington, DC , USA
Farke Andrew
Electronic publication date: 2015 Jun 9
Publication date: 2015
Volume: 3
Electronic Location ID: e986
Received 2014 Dec 15; Accepted 2015 May 8
Copyright: © 2015 Pittman et al.
Copyright year: 2015
Copyright holder: Pittman et al.
License: This is an open access article distributed under the terms of the Creative Commons Attribution License, which permits unrestricted use, distribution, reproduction and adaptation in any medium and for any purpose provided that it is properly attributed. For attribution, the original author(s), title, publication source (PeerJ) and either DOI or URL of the article must be cited.
License URL: https://creativecommons.org/licenses/by/4.0/

Keywords: Alvarezsauroid, Parvicursorine, Theropod, Upper Cretaceous, Campanian, Inner Mongolia

Funding: National Natural Science Foundation of China 41120124002 973 (National Basic Research) program 2012CB821900 Department of Land and Resources Faculty of Science of the University of Hong Kong EAPSI fellowship 1311000 Robert Weintraub Fellowship This work was supported by the National Natural Science Foundation of China (41120124002), 973 (National Basic Research) program (2012CB821900) and the Department of Land and Resources, Inner Mongolia, China. MP’s participation in the 2013 expedition was funded by the Faculty of Science of the University of Hong Kong. JS’s participation in the expedition was funded by a United States National Science Foundation East Asia and Pacific Summer Institutes (EAPSI) fellowship (1311000). Research by JS was also supported by the Robert Weintraub Fellowship in Systematics and Evolution (George Washington University). The funders had no role in study design, data collection and analysis, decision to publish, or preparation of the manuscript.

==============================
A new parvicursorine alvarezsauroid theropod specimen IVPP V20341 from the Upper Cretaceous Wulansuhai Formation of Bayan Mandahu, Inner Mongolia, China is described. IVPP V20341 appears to be distinguishable amongst alvarezsauroids by possible cervical procoely and relatively larger semi-circular caudal neural canals, but these features are not proposed as autapomorphies because current knowledge of alvarezsauroid necks and tails remains sparse. IVPP V20341 is distinguishable from Linhenykus—the sole parvicursorine at Bayan Mandahu—by the location of the origination points of the anterior caudal transverse processes; in IVPP V20341 this is the anterodorsal corner of the centra, whereas in Linhenykus it is the posterior end of the prezygapophyses. A number of additional tentative differences between IVPP V20341 and Linhenykus are also identified, but cannot be confirmed until further details of anatomical variation along the neck and tail are revealed by future finds. Thus, following the study of IVPP V20341 there are still seven parvicursorine species from the Upper Cretaceous Gobi Basin, but future finds could increase this to eight species.

Introduction

The Upper Cretaceous rocks of the Gobi Basin of China and Mongolia have yielded alvarezsauroid theropods with impressive specialised body plans, including the uniquely monodactyl parvicursorine Linhenykus monodactylus (Xu et al., 2010b). The latter is the only parvicursorine species from the Chinese Gobi Basin and was discovered in Bayan Mandahu, Inner Mongolia. In contrast, six parvicursorine species are known from the Mongolian Gobi Basin: Albinykus (Nesbitt et al., 2011), Ceratonykus (Alifanov & Barsbold, 2009), Kol (Turner, Nesbitt & Norell, 2009), Mononykus (Perle et al., 1994; Perle et al., 1993), Parvicursor (Karhu & Rautian, 1996; Longrich & Currie, 2009) and Shuvuuia (Chiappe, Norell & Clark, 1998; Suzuki et al., 2002) (Table S1). Dinosaur discoveries at Bayan Mandahu have been important in demonstrating that distinct faunas existed across the Upper Cretaceous Gobi Basin (Xu et al., 2010b; Longrich & Currie, 2009; Xu et al., 2012; Godefroit et al., 2008; Longrich, Currie & Dong, 2010; Xu et al., 2010a; Xu et al., 2015; Xu et al., 2013a), which has provided valuable insight into how dinosaurs behaved and coped over expansive semi-arid environments. Here we report IVPP V20341 a fragmentary disarticulated parvicursorine specimen that was discovered in Bayan Mandahu during the 2013 field season of the Inner Mongolia Research Project (IMRP). IVPP V20341 appears to have two autapomorphies—possible cervical procoely and relatively larger semi-circular caudal neural canals. However, these cannot be confidently assigned because the large amounts of missing data from known parvicursorine neck and tail specimens makes it possible that these features could be present in more complete future specimens of existing taxa. IVPP V20341 is therefore an important parvicursorine specimen as comparative studies with future finds may either provide new information about anatomical variation in these animals or justify the erection of a new taxon, the second and eighth parvicursorine of Bayan Mandahu and the Gobi Basin respectively.

Systematic Palaeontology

Dinosauria Owen, 1842	
Theropoda Marsh, 1881	
Alvarezsauridae Bonaparte, 1991	
Parvicursorinae Karhu & Rautian, 1996	

Material

IVPP V20341, a highly fragmentary postcranial skeleton comprising of: an articulated series of 4 partial anterior cervical vertebrae, an isolated anterior portion of an anterior cervical vertebra; 4 isolated fragmentary anterior caudal vertebrae; a potential partial left scapula; various possible pedal phalanges, including unknown digit II/III and IV phalanges, a right II-1, II-2, and IV, as well as a potential left III-2 and III-3.

Locality and horizon

IVPP V20341 was found at Bayan Mandahu, Inner Mongolia, China, a locality of the Campanian-aged Wulansuhai Formation (Eberth, 1993). On July 2nd 2013, a team member (JBS) discovered isolated alvarezsauroid bones weathering out of a clifftop exposure of a fine-grained, red structureless aeolian quartz arenite, ∼3 km SE of ‘The Gate’ locality (41°43′15.3″N, 106°44′43.3″E; Fig. 1), close to the location of Eberth’s (1993) ‘7/12/90/2’ stratigraphic section, but not as far North as his ‘7/12/90/1’ section (Eberth, 1993). The locality lies within an larger area that Jerzykiewicz et al. (1993) called the ‘South Escarpment’ locality, but the specific locality of IVPP V20341 is referred to as the ‘Eagle’s Nest’ because a large predatory bird nest was found ∼3 m from the find. On July 3rd 2013, another team member (MP) further explored the sublocality and recovered several additional bones within 1 m of the original material. Following the depositional environments and facies transitions identified by Eberth (1993) at Bayan Mandahu, the specimen was deposited in the zone 1 palaeographic zone. The latter consists of alluvial, lacustrine and aeolian sediments deposited in a distal alluvial fan or braid-plain environment adjacent to an aeolian dune field. Given the sandy depositional environment of the specimen, IVPP V20341 was nicknamed ‘Xiaoshalong’, which is Mandarin Chinese for ‘little sand dragon’.

Figure 1 IVPP V20341 locality.

Place of discovery for IVPP V20341 (41° 43′15.3″N, 106° 44′43.3″E), ∼3 km SE of ‘The Gate’ locality and close to the location of Eberth’s ‘7/12/90/2’ stratigraphic section (Eberth, 1993) (After Jerzykiewicz et al., 1993; Eberth, 1993).

Description and comparison

IVPP V20341 appears to belong to an individual at or near skeletal maturity (e.g., subadult or adult) because the neurocentral sutures on the vertebrae preserved appear to be completely closed (Brochu, 1996; Irmis, 2007). However, this inference should be treated as a tentative one in the absence of other ontogenetically-informative fusion in the appendicular skeleton as well as relevant histological data. The skeletal elements that are common to both IVPP V20341 and Linhenykus are generally smaller in the former than in the latter. This suggests that IVPP V20341 was probably lighter in weight than Linhenykus, which itself weighed around 450 g (Xu et al., 2010b; Xu et al., 2013b). IVPP V20341 lacks a femur and skull so a more accurate proxy-based body mass estimate was not possible (Christiansen & Fariña, 2004; Therrien & Henderson, 2007). The incomplete vertebral column and the missing skull also prohibited a meaningful measurement of body length. However, given the relative size of IVPP V20341 to Linhenykus, the former was probably lighter than other Asian parvicursorines, with the exception of Shuvuuia and Parvicursor (see Table S2).

Axial skeleton

An articulated series of 4 partial anterior cervical vertebrae (A–D) (Fig. 2A), an isolated anterior portion of an anterior cervical vertebra (Fig. 2B) and 4 isolated fragmentary anterior caudal vertebrae (A–D) represent the axial skeleton (Fig. 3).

Figure 2 Cervical vertebrae of IVPP V20341.

(A) dorsal, ventral, left lateral and right lateral views of an articulated series of four partial anterior cervical vertebrae (cervicals A–D). (B), anterior, right lateral, dorsal and ventral views of an isolated anterior portion of an anterior cervical vertebra. Abbreviations: cc af, concave articular face; cc vr, concave ventral rim of the anterior articular face; cvx vr, convex ventral rim of the posterior articular face; cv, cervical vertebra; cv r?, cervical vertebra rib?; di, diapophysis; dr, diapophyseal ridge; lc ls, laterally compressed lateral surface; nc, neural canal; ns, neural spine; poz, postzygapophysis; prz, prezygapophysis; r, ridge; r vs, rounded ventral surface. Scale = 5 mm.

Figure 3 Caudal vertebrae of IVPP V20341.

Anterior caudal vertebrae (A–D) in anterior, posterior, lateral and ventral views (caudal D is damaged in anterior view so its dorsal view is shown instead). Abbreviations: cc af, concave articular face; ch?, possible chevron; cx af, convex articular face; fo, foramina; ns, neural canal; ns, neural spine; poz, postzygapophysis; prz, prezygapophysis; r, ridge; tp, transverse process; vf?, potential ventral furrow. Scale = 5 mm.

Cervical vertebrae

The first of four articulated partial cervicals (cervical A) is highly fragmentary whereas the second cervical in the series (cervical B) is almost completely preserved save for a small degree of dorsal crushing and abrasion. The third cervical in the preserved series (cervical C) is partially complete and is best represented on its left lateral side. The most posterior cervical preserved (cervical D) is also highly fragmentary, as in cervical A. An isolated cervical centra resembling the anteroventral portion of a smaller version of cervical B is preserved. However, the dorsal surface—including most of the neural arch—is missing. Linear measurements of the cervical vertebrae are given in Table 1.

Table 1 Vertebral measurements.

Selected measurements of the cervical and caudal vertebrae preserved.

Dimension in mm	Vertebral element	
	Cv A	Cv B	Cv C	Cv D	Isolated cervical	Cd A	Cd B	Cd C	Cd D	
Anteroposterior length between the dorsal rim of the anterior and posterior articular surfaces	–	6.89 (b)	–	–	–	6.70 (l)	5.26 (b, l)	–	–	
Anteroposterior length between the dorsal rim of the anterior articular surface and the posteriormost tip of the posterior articular condyle	–	–	–	–	–	7.53 (l)	7.29 (l)	–	–	
Lateral width of the anterior articular face	–	4.20	–	–	4.24	3.87 (b)	4.12 (b)	4.34 (b)	–	
Lateral width of the posterior articular face (at the rim of the articular surface)	4.88 (b)	–	–	–	–	3.86	4.05	–	3.84	
Centrum height (dorsoventral height between the ventral and dorsal rims of the posterior articular surface)	–	1.87 (b)	–	–	3.07 (a)	3.26 (l)	3.13 (l)	3.53 (a)	3.21 (l)	
Prezygapophyseal angle from the vertical in ° (anterior view)	–	–	78 (l)	64 (l)	–	26 (l)	–	26 (l)	–	
Postzygapophyseal angle from the vertical in ° (posterior view)	–	50 (l)	40 (b, l)	–	–	76 (b, r)	–	–	–	
Neural spine height (dorsoventral height between the dorsal rim of posterior articular surface and the neural spine tip)	–	5.89 (b, l)	–	–	–	4.79 (l)	–	–	–	
Notes.

b partial/damaged/matrix-obscured feature resulting in underestimated dimensions and approximate angles

l left lateral side

r right lateral side

a anterior portion available only

Cervicals A and B in IVPP V20341—a cervical fragment and a dorsally crushed complete cervical respectively—meet via a possible procoelous articular joint that is identified from the shape of the ventral rims of the articular faces: in cervical A, the posterior articular face has a slightly abraded convex ventral rim, whereas in cervical B, the anterior articular face has a well-preserved concave ventral rim (Fig. 2A). However, cervical procoely in IVPP V20341 cannot be confirmed unequivocally until a disarticulated procoelous cervical is discovered since cervicals A and B could not be safely separated from one another during preparation. Nevertheless, this possible cervical procoely is unlike the strong cervical opisthocoely found in Linhenykus, Shuvuuia and Mononykus (Xu et al., 2010b; Perle et al., 1994; Xu et al., 2013b; Chiappe, Norell & Clark, 2002). Cervical opisthocoely has also been proposed—albeit tentatively—for Patagonykus (Chiappe, Norell & Clark, 2002; Novas, 1996; Novas, 1997). These articular joint morphologies contrast to the amphi- to platycoelous condition in the basalmost alvarezsauroid Haplocheirus (Choiniere et al., 2010) and the amphiplatyan condition proposed in Alvarezsaurus (MUCPv 54; Chiappe, Norell & Clark, 2002; Bonaparte, 1991). Despite the prevalence of opisthocoely amongst alvarezsauroid cervicals, the paucity of complete alvarezsauroid cervical series (one complete neck for the basal alvarezsauroid Haplocheirus solers (IVPP V15988) and two near complete ones for Mononykus (MPC 107/6; Perle et al., 1994) and Shuvuuia (MPC 100/975; Chiappe, Norell & Clark, 2002)) warrants caution in considering cervical procoely as a potential autapomorphic characteristic of IVPP V20341, particularly given the array of articular face geometries that are preserved in the dorsal and caudal vertebral series of other alvarezsauroids e.g., the opisthocoelous proximal and mid-dorsals and biconvex distal dorsals of Mononykus (MPC 107/6; Perle et al., 1994) and the procoelous, amphicoelous and opisthocoelous/biconvex proximal caudals of Achillesaurus (MACN-PV-RN 1116; Martinelli & Vera, 2007). Thus, the possible cervical procoely observed in IVPP V20341 is considered as an equivocal autapomorphy that cannot be used to erect a new taxon on its own for the aforementioned reasons. As in other alvarezsauroids, the condyles of the cervical centrum preserved appear smaller than their corresponding articular surfaces (Xu et al., 2013b). In ventral view, the rims of the anterior articular surfaces are concave whilst the posterior ones are convex.

In lateral view, the cervical centra of IVPP V20341 are long and low, as in other alvarezsauroids (Chiappe, Norell & Clark, 2002). They are not as strongly laterally compressed as the cervicals of Linhenykus (IVPP 17608; Xu et al., 2013b) as only the ventral portion is compressed in IVPP V20341. Thus, the lateral surfaces of the cervicals of IVPP V20341 are more vertical in the middle portion of the centrum than in Linhenykus, although the former cervicals may not be in analogous positions to those in Linhenykus. Nevertheless, the specimen’s laterally compressed cervicals qualified it as an alvarezsaurid alvarezsauroid (cervical centra bearing deep lateral depressions is an alvarezsaurid synapomorphy (Xu et al., 2013b)—character state 8.1 of Longrich & Currie (2009)).

The ventral surfaces of these cervicals are rounded and slightly pinched along their mid-length whereas they are grooved (longitudinal ventral furrow) and pinched along their mid-length in the cervicals of Linhenykus, IVPP V17608 (Xu et al., 2013b). Shuvuuia appears to share the latter morphology in its posterior cervical vertebrae (MPC 100/975; Chiappe, Norell & Clark, 2002) and Ceratonykus appears to as well (MPC 100/124; Alifanov & Barsbold, 2009; a furrow is present in the cranioventral and midventral positions of a posterior cervical, but the posteroventral portion is partially missing). In the most anterior cervicals of Shuvuuia the ventral furrow does not span the entire centra, because the mid-point of the centra is interrupted by a rounded surface (MPC 100/975; Chiappe, Norell & Clark, 2002), as in Mononykus (MPC 107/6; Perle et al., 1994). The cranioventral furrows in Shuvuuia and Mononykus are bordered by prominences (Chiappe, Norell & Clark, 2002) and are the only furrows present in the most anteriorly preserved cervicals of Shuvuuia. The presence of a full length ventral furrow in posterior centra of Shuvuuia and Ceratonykus suggests that this feature is probably not a valid autapomorphy of Linhenykus (Xu et al., 2013b), unless future data can demonstrate that only Linhenykus has this furrow on all of its cervicals. The lack of a prominence-bordered cranioventral furrow in cervical B of IVPP V20341 appears unique to Asian parvicursorines, but it is known in South American alvarezsauroids. MCF-PVPH 38, a possible fragmentary 5th cervical from an indeterminate Argentine alvarezsauroid (?Alvarezsauridae indet.), has a straight, narrow and rounded ventral surface—much wider than a keel—with a ‘veiny’ surface texture (Novas, 1997). However, the cervicodorsals of Mononykus (MPC 107/6; Perle et al., 1994) have rounded ventral surfaces too so cervical B of IVPP V20341 may well represent a posterior rather than anterior cervical.

Dorsal to the left lateral postzygapophysis of cervicals B and C there is no evidence of an epipophysis, indicating that IVPP V20341 is an alvarezsauroid theropod (character state 6.1 of Longrich & Currie (2009) is an alvarezsauroid synapomorphy). In IVPP V20341, the diapophyseal ridge has a convex profile in the area around the small nubbin-like diapophysis, but shallows gradually towards the posteroventral corner of the centrum. This differs from the condition in Linhenykus where the diapophyseal ridge extends to the posterodorsal rim of each middle cervical centra—another autapomorphy of this taxon (Xu et al., 2013b). However, the latter may not be a valid autapomorphy owing to the presence of the same feature in Shuvuuia (MPC 100/975; Chiappe, Norell & Clark, 2002). The relationship between the posterior termination point of the diapophyseal ridge and the position of its respective cervical along the neck is unclear. Thus, it is plausible that the differences in the condition between IVPP V20341 and both Linhenykus and Shuvuuia could be an artefact of cervical position. In IVPP V20341 the diapophyseal ridge’s anteroventral surface is excavated and houses a broad shallow fossa. This feature is difficult to appraise in Linhenykus as the aforementioned ridge and anteroventral surface are not preserved in the same cervical owing to specimen damage. However, across two middle cervicals large collateral pneumatic foramina are present (Xu et al., 2013b), instead of broad shallow fossa.

Cervical B lacks a carotid process unlike the cervicals of Linhenykus (where it is confluent with the anterior ends of the ventral ridges (Xu et al., 2013b)), the anterior cervicals of Shuvuuia (Chiappe, Norell & Clark, 2002) and Mononykus (Perle et al., 1994) as well as the cervicals of other theropods, including some ornithomimosaurs, oviraptorosaurs and paravians (Xu et al., 2013b). Cervicals B and C of IVPP V20341 both lack pneumatic foramina as in Mononykus (Perle et al., 1994), but the lateral surfaces of their centra appear to be less mediolaterally compressed than in Mononykus (mediolaterally compressed cervical centra that lack pneumatic foramina are given in the diagnosis of Mononykus (Chiappe, Norell & Clark, 2002)). In Linhenykus—as in Alvarezsaurus and Shuvuuia—pneumatic foramina occupy the area immediately posterior to the parapophyses (Xu et al., 2013b; Chiappe, Norell & Clark, 2002; Bonaparte, 1991). However, this difference might be explained by positional discrepancies between cervicals B and C of IVPP V20341 and the cervicals of Linhenykus. As in Linhenykus, the neural pedicles are mediolaterally broad and dorsoventrally low and it appears that the anterior edge of each pedicle is also flush with the anterior articular surface of the centrum, whereas the posterior edge is anterior to the posterior articular surface (excluding the condyle) (Xu et al., 2013b). The parapophyses are also low, laterally projecting eminences as in Linhenykus (Xu et al., 2013b). The process is dorsolaterally orientated.

The zygapophyseal articular facets in cervicals B–D of IVPP V20341 have low-angles (∼78° and ∼61° from the vertical (in anterior view) for the prezygapophyses of cervicals C and D respectively and ∼50° and ∼40° from the vertical (in posterior view) for the postzygapophyses of cervicals B and C respectively) suggesting a greater range of motion in the horizontal plane than the vertical one. This is because the prezygapophyses show the latter; whilst the postzygapophyses are complimentary to this pattern since they indicate that the range of motion was similar in either plane. The prezygapophyses are anteroposteriorly short and extend over approximately one third of the preceding centra. The postzygapophyseal processes of cervical B of IVPP V20341 are separated by a wider angle (∼136°) in comparison to the middle cervicals preserved in Linhenykus (∼105°). However, this difference may simply reflect differences in anatomical position so should be treated with caution. The postzygapophyses (left one on cervical B) appear to be dorsally orientated, as in Linhenykus (Xu et al., 2013b). In dorsal view the postzygapophyses have a nearly straight medial edge and a convex lateral edge, as in other Asian alvarezsauroids (Xu et al., 2013b; Chiappe, Norell & Clark, 2002). This contrasts with the postzygapophyses of Alvarezsaurus, which have convex medial and lateral edges that create a paddle-like shape in dorsal view (Bonaparte, 1991). Epipophyses are absent from the postzygapophyses (as evident from the left lateral sides of cervicals B and C), unlike the mid-cervicals of Linhenykus, which have weakly developed ridge-like ones that are an autapomorphy of this taxon (Xu et al., 2013b). In IVPP V20341 the prezygapophyses are more widely separated laterally and have larger articular surfaces in comparison to the postzygapophyses—this pattern is not observed in Linhenykus, which has laterally narrower prezygapophyses than postzygapophyses in the middle cervicals that are preserved (Xu et al., 2013b). In Shuvuuia (MPC 100/975) the prezygapophyses are laterally wider than the postzygapophyses in the proximal proportion of the cervical series, have a similar lateral width to the postzygapophyses in the mid-series (at a currently undefined transition point due to the incompleteness of the cervical series), whilst in the distal portion of the series the prezygapophyses are laterally narrower than the postzygapophyses (Chiappe, Norell & Clark, 2002) (as in Linhenykus (Xu et al., 2013b)). If this pattern of zygapophyseal width is similar in other parvicursorines it suggests that cervicals A–D are anterior ones. This implies that the presence of mid-cervical epipophyses in IVPP V20341—as in Linhenykus (Xu et al., 2013b)—is uncertain, since mid-cervicals are not preserved in IVPP V20341. The suggestion that cervicals A–D are anterior ones is consistent with the partially-damaged neural spine observed in cervical B, which rises from a well-defined dorsal ridge—that spans the whole anterior dorsal surface of the vertebra—at the approximate position of the postzygapophyseal facets. Taking into account the damage to this neural spine it appears to be dorsoposteriorly directed, but it would be speculative to comment on both its dorsoventral height and anteroposterior length in relative terms. The anterior position of cervicals A–D along the neck is at odds with the posterior position of them that is suggested by the similarity of their rounded ventral surfaces to the cervicodorsals of Mononykus (MPC 107/6; Perle et al., 1994). However, this conflict appears to be resolved by the presence of a rounded ventral surface in the 5th cervical of MCF-PVPH 38 which suggests that the latter is a feature of the base of the neck. This condition is not observed in the anterior cervicals of Shuvuuia (MPC 100/975; Chiappe, Norell & Clark, 2002) and Mononykus (MPC 107/6; Perle et al., 1994). However, it is unclear if the same condition is present in Linhenykus as even if its preserved cervicals included anterior ones—as suspected by Xu et al. (2013b)—their incompleteness means that they could potentially possess the condition discussed. The anterior position of cervicals A–D along the neck also means that the lateral surfaces of their centra are not directly comparable to the lateral surfaces of the middle cervicals of Linhenykus. The incompleteness of these cervical series therefore makes it premature to suggest that a given lateral surface morphology is unique to their respective series. Similarly, the absence of a carotid process on cervical B cannot be directly compared to the carotid process present on the cervicals of Linhenykus because both morphologies may be present in more complete neck specimens. The diapophyseal ridge morphology of the anterior cervicals of IVPP V20341 is significantly different from the morphology in the middle cervicals of Linhenykus. However, it is uncertain if these morphologies are transitional along the anterior and middle portions of the neck. Similarly, it is unclear if the absence or presence of pneumatic foramina relates to differences in cervical position between the preserved portions of IVPP V20341 and Linhenykus owing to the large number of missing cervicals, even though the anteriormost cervicals preserved in Linhenykus may actually be anterior ones (IVPP 17608; Xu et al., 2013b).

One possible cervical rib is preserved in association with the posterior portion of cervical A. This element is identified as such because of its long, thin shape and its association with a cervical. However, breakage in this element—especially anteriorly—means that this identification is equivocal. Even so, there is no evidence that seems to support the fusion of the cervical ribs with their associated vertebrae, unlike in Shuvuuia (IGM 100/977; Chiappe, Norell & Clark, 2002).

The partial isolated cervical vertebra resembles a smaller version of cervical B based on the anterior portion that is preserved. Therefore as in cervical B, this cervical was probably procoelous (posterior articular surface is missing), and from the small portion that is preserved, it seems to have a smooth ventral surface that is pinched away from the anterior edge of the centrum. The smaller size of the isolated cervical relative to cervical B potentially suggests a more posterior position along the series than the latter (more posterior anterior cervical?).

The neural canal is poorly exposed in cervicals A–D, but the anterior portion of the canal is fully exposed in the isolated cervical, owing to its largely missing neural arch. The latter neural canal is proportionally larger compared to other vertebra, as in most alvarezsauroids (Chiappe, Norell & Clark, 2002). It appears to slope downwards in a posteroventral direction and has a mid-line ridge along its ventral surface.

Caudal vertebrae

IVPP V20341 includes four isolated anterior caudal vertebrae: one that is well-preserved and three others that are poorly preserved (Fig. 3). These are referred to as caudals A–D in order of their relative anteroposterior position along the tail, as determined using the anteroposterior position of the neural arch pedicle and transverse processes and the relative development of the furrows and ridges along the ventral surfaces of the centra. The dimensions of the caudal vertebrae given in Table 1 do not unequivocally support the proposed ordering, nor any other ones. This probably reflects the large amount of missing data, particularly in caudals C and D (Table 1), and variability in vertebral geometry changes along the tail, as has been measured in a wide range of theropods (Pittman et al., 2013).

Caudals A–C appear to be anterior ones because the neural arch pedicle is anteriorly placed along the anteroposterior length of the centrum, as evident in other parvicursorine alvarezsauroids including Alvarezsaurus (MUCPv 54; Chiappe, Norell & Clark, 2002; Bonaparte, 1991), Achillesaurus (MACN-PV-RN 1116; Martinelli & Vera, 2007), Linhenykus (IVPP V17608; Xu et al., 2010b; Xu et al., 2013b), Parvicursor (PIN 4487/25; Karhu & Rautian, 1996), Shuvuuia (MPC 100/975; Chiappe, Norell & Clark, 2002) and Xixianykus (XMDFEC V0011; Xu et al., 2010c).

In lateral view the posterior margin of the narrowest portion of caudal A’s neural arch (the neck) is approximately two-thirds along the anteroposterior length of the centrum, whereas in the first free caudal of Parvicursor (PIN 4487/25; Karhu & Rautian, 1996) and the proximal caudals of Alvarezsaurus (MUCPv 54; Chiappe, Norell & Clark, 2002; Bonaparte, 1991) this is less than halfway along the same length and in the first caudal of Xixianykus (XMDFEC V0011; Xu et al., 2010c) this is about halfway along the same length. The condition in IVPP V20341 is comparable to the supposed first caudal of Patagonykus (MCF-PVPH 37; Novas, 1997) and the middle and distal caudals of Alvarezsaurus (Chiappe, Norell & Clark, 2002; Bonaparte, 1991). This suggests that the position of the posterior margin of the neural arch neck along the anteroposterior length of the caudal centrum of IVPP V20341 is unhelpful in determining the relative position of caudals A–C along the tail.

Caudals A to D show a posterior migration of the transverse processes (only the posterior ridge of the process is visible on the left lateral side of caudal D), but these are all still situated anteriorly on the centrum, which identifies IVPP V20341 as a parvicursorine alvarezsauroid (Xu et al., 2013b). The transverse processes form a broad subhorizontal ridge that originates from the anterodorsal corner of the centra (caudals A and B) rather than the posterior end of the prezygapophyses as in Linhenykus (best examples in caudals 3, 5, 7 and 8 of IVPP V17608; Xu et al., 2013b) and Shuvuuia (anterior caudals (caudals 3–8?) of MPC 100/975; Chiappe, Norell & Clark, 2002). It may be possible that caudals A and B of IVPP V20341 are caudals 1 and 2 of Linhenykus, but this is unlikely given that the first caudal of the latter has a sharp ventral keel (Xu et al., 2013b), a feature that neither caudals A or B possess. In both IVPP V20341 and Linhenykus the broad subhorizontal ridge does deflect posteriorly towards the dorsal edge of the posterior articular face (Xu et al., 2013b), although this appears to deflect more ventrally in the former specimen. However, the latter probably relates to subtle differences in caudal position between these specimens.

The caudals become anteroposteriorly shorter from caudal A to B (caudals C and D are not anteroposteriorly complete), but this pattern is not emphasised here because centrum length does not decrease continuously along the tail of Linhenykus—its middle caudals are actually anteroposteriorly longer than its anteriormost ones (IVPP V17608; Xu et al., 2013b). Caudals A and B possess a longitudinal furrow along the centrum’s entire ventral surface and this is bordered laterally by two ventral keels. This feature is also observed in Linhenykus (Xu et al., 2013b), Parvicursor (PIN 4487/25; Karhu & Rautian, 1996; Chiappe, Norell & Clark, 2002), Patagonykus (supposed 20th caudal of MCF-PVPH 37; Novas, 1997) and Shuvuuia (MPC 100/975; Chiappe, Norell & Clark, 2002). However, this feature is less developed in caudal B and is barely visible in caudal C, where the ridges are low and the furrow is broad and shallow. The ventral surface of caudal D has been eroded down to the cortical bone. The ventral surfaces of caudals A, B and C therefore support their proposed positional ordering: caudal A is the anteriormost one whilst caudal C is the posteriormost one.

Caudals A–D are laterally pinched. Caudals A and B are procoelous, but this cannot be confirmed in caudals C and D owing to a missing posterior articular face in the former and a missing anterior articular face in the latter. In caudals A and B, the concave anterior articular face is deep, whilst the posterior condyle is well-developed and hemi-spherical in shape. Procoely in caudal vertebrae is also observed in Haplocheirus (IVPP V15988; Choiniere et al., 2010), Shuvuuia (MPC 100/975; Chiappe, Norell & Clark, 2002), Mononykus (MPC N107/6; Perle et al., 1994), Xixianykus (XMDFEC V0011; Xu et al., 2010c), Alvarezsaurus (MUCPv 54; Chiappe, Norell & Clark, 2002; Bonaparte, 1991) and potentially in Patagonykus (MCF-PVPH 37; only the posterior articular surfaces are preserved (Novas, 1997)). However, the first caudal of Linhenykus is amphiplatyan (Xu et al., 2013b) whilst a proximal caudal of Achillesaurus—tentatively assigned as the fourth in the series—is amphicoelous (biconcave) (Martinelli & Vera, 2007).

Caudals A–D lack the sharp ventral keel that has been associated with the anteriormost caudals of many parvicursorines, so they probably represent caudals located further along the tail—at least posterior to the ventrally-keeled first caudal, as in Linhenykus and Xixianykus (Xu et al., 2013b; Xu et al., 2010c). Shuvuuia and Achillesaurus have a sharp ventral keel on their first two caudals (the first two caudals of Shuvuuia as identified by Chiappe, Norell & Clark (2002); a keel is not observed on the first caudal of Achillesaurus owing to specimen damage, but this feature is inferred because of the keels present on the last sacral and second caudal (Martinelli & Vera, 2007)) and this feature is also present on an anterior caudal of Alvarezsaurus (Bonaparte, 1991), Mononykus (Perle et al., 1994) and, supposedly, of Parvicursor as well (Karhu & Rautian, 1996). Patagonykus has a seemingly unique ventral surface amongst parvicursorines as the assumed first caudal has a ventral surface that is transversely narrow and slightly flat (Novas, 1996).

No evidence of chevron articulation facets was found on caudals A, B and D—the only ones that preserve the posterior ventral surface—as in caudals 1–3 of Linhenykus (IVPP V17608). However, these facets are well-developed on the posteroventral surface of a distal caudal—supposedly the 20th caudal—of P. puertai (MCF-PVPH 37; Novas, 1997). In Linhenykus (IVPP V17608) chevron articulation facets are weakly developed on the posteroventral surfaces of more posterior proximal caudals (caudals 4 and 5) and strongly developed on the anteroventral surface of a middle caudal (caudal 13) (Xu et al., 2013b). However, the absence of chevron articulation facets on caudals A, B and D of IVPP V20341 could become a distinguishing feature from Linhenykus if further finds can demonstrate that this morphology exists on or beyond caudal 4, in contrast to Linhenykus.

On the anterior portions of the right and left lateral surfaces of caudal centra A and B respectively, there is a very small weakly developed foramen, but this is absent on the opposing side of the centrum. However, the small size of these foramen and their differing positions suggests that they probably have little taxonomic significance. The lateral surfaces of caudals A–D lack both large, oval-shaped and small, subcircular fossa—unlike the first and second caudals of Patagonykus (MCF-PVPH 37; Novas, 1996) and Achillesaurus (MACN-PV-RN 1116; Martinelli & Vera, 2007) respectively—further supporting that caudals A–D are not the anteriormost ones. Foramen are absent from the caudals of Linhenykus but IVPP V20341 and Linhenykus (IVPP V17608) both have broad, shallow fossa on the lateral surfaces of their centra (e.g., caudal 5 and caudals A–D respectively).

The neural spine of caudal A is partially preserved and is missing its dorsoposterior portion. However, with what is present it is evident that the neural spine is rod-like, quite tall dorsoventrally, anteroposteriorly short and dorsoposteriorly directed. This suggests that caudal A is an anterior one as this type of neural spine morphology is found in the proximal caudals of Linhenykus (caudal two of IVPP V17608; Xu et al., 2013b), Parvicursor (the neural spine of the first caudal in PIN 4487/25 is dorsoventrally tall and dorsoposteriorly directed overall, but its rounded tip protrudes by a relatively small height beyond the dorsal margin of the postzygapophyseal facets and is dorsally directed (Karhu & Rautian, 1996)), Patagonykus (the first caudal of MCF-PVPH 37 is dorsoventrally tall and weakly dorsoposteriorly directed (Novas, 1997)) and Shuvuuia (MPC 100/975; Chiappe, Norell & Clark, 2002). In contrast, the proximal neural spines of Alvarezsaurus (MUCPv 54) are dorsally directed and have a subtriangular lateral profile (Chiappe, Norell & Clark, 2002). The anterior margin of the damaged neural spine on caudal A lies above the neural pedicle, whereas the whole neural spine is located posterior to the pedicle—at the level of the postzygapophyses —in the anterior caudals of Linhenykus (e.g., caudals 2–4 of IVPP V17608; Xu et al., 2013b). This difference is probably because of the differing caudal positions being compared, given that theropod neural spines generally extend further posteriorly as their inclination increases along the anterior portion of the tail. However, caudal A of IVPP V20341 does not appear to be an anteriormost caudal because of the absence of a sharp ventral keel and the presence of procoelous articular faces and fossae, as mentioned earlier in the caudal vertebrae section. This suggests that the anterior and posterior margins of caudal A’s neural spine are perhaps more damaged than they seem to be.

The shape of the neural canals in caudals A–C are laterally wider and more semi-circular compared to the laterally narrower and more oval-shaped ones of Linhenykus (observable in caudals 2, 7 and 13 of IVPP V17608; Xu et al., 2013b), Patagonykus (MCF-PVPH-37, the supposed first caudal (Novas, 1997)) and Parvicursor (PIN 4487/25, the supposed first caudal (Karhu & Rautian, 1996)). However, this feature cannot be confirmed as an autapomorphy of IVPP V20341 because it is plausible that laterally wider and more semi-circular neural canals might be present in as yet unknown portions of the vertebral columns of existing parvicursorines, since neural canal size and shape changes along the vertebral column of theropods (and other vertebrates). The ventral surface of the neural canal of caudal D bears a longitudinal ridge. However, the distribution of this characteristic amongst parvicursorines is unclear owing to a paucity of appropriate specimens.

Appendicular skeleton

The appendicular skeleton comprises of a potential left scapular shaft (Fig. 4) and a range of possible pedal phalanges (Fig. 5), including unknown digit II/III and IV phalanges, a right II-1, II-2 and IV, as well as a potential left III-2 and III-3. Linear measurements of the appendicular skeleton are given in Table 2.

Figure 4 Scapula of IVPP V20341.

Scapular blade in dorsal and posterior views. Scale = 5 mm.

Figure 5 Hindlimb elements of IVPP V20341.

Pedal phalanges possibly with the identities: (A) ?right II-1, (B) ?right II-2, (C) II/III, (D) ?left III-2, (E) ?left III-3, (F) ?right IV, and (G) ?IV. Abbreviations: af, articular facet; c, cavity; icg, intercondylar groove; lf, ligamental fossae; m co, medial condyle; ov, overhang; p, prominence; r, ridge. Scale = 5 mm.

Table 2 Appendicular element measurements.

Selected measurements of elements from the appendicular skeleton, including estimated ones.

	Vertebral element	
	MTIII (right)	II-1 (right)	III-2 (right)	III-3 (left)	
Maximum anterior articular surface dorsoventral height	–	2.92	–	2.33	
Maximum anterior articular surface lateral width	–	2.72	–	2.45	
Maximum posterior articular surface dorsoventral height	3.54(b)	–	3.07	–	
Maximum posterior articular surface lateral width	3.56(b)	–	2.71	–	
Maximum anteroposterior length	–	–	–	6.07(b)	
Notes.

b partial/damaged resulting in underestimated dimensions

Forelimb

Left scapular shaft

In the same small block of sediment that contains cervicals A–D there is a partial, strap-like piece of bone (Fig. 4). The most complete margin of this bone is deflected and at one end of the bone (distal end?) the margins are subparallel. At the presumed distal end of this bone the generally flat surface sinks into two shallow grooves that traverse towards the midline to create a flattened triangular eminence. The bone superficially resembles an alvarezsauroid scapular shaft (preserved in Haplocheirus (IVPP V15988; Choiniere et al., 2010), Bonapartenykus (MPCA 1290; Agnolin et al., 2012), Alvarezsaurus (MUCPv 54; Chiappe, Norell & Clark, 2002), Mononykus (MPC 107/6; Perle et al., 1994) and Shuvuuia (MPC 100/977); Table S1), but it does not preserve enough information to help differentiate it amongst alvarezsauroids save for the triangular eminence. However, the latter feature has not been reported in the aforementioned alvarezsauroids so this could be a distinguishing feature of IVPP V20341 if this bone fragment is indeed part of a scapula (potentially from the left side of the body).

Hind limb

Possible right pedal phalanx II-1

The distal end of a possible pedal phalanx II-1 is preserved (Fig. 5A). This has distinct condyles that are separated by an intercondylar groove, but the latter is narrower and less developed than in preserved phalangeal elements of Linhenykus (IVPP V17608; Xu et al., 2013b: right manual phalanx II-1; left pedal phalanges I-1, I-2, II-1 and IV-1, ?right pedal phalanges II-1–II-3 and IV-4, ?left pedal phalanges IV-3–IV-5; IVPP V18190 (Hone et al., 2013): left pedal phalanges III-1, IV-1 and IV-2), Mononykus (MPC 107/6; Perle et al., 1994: complete set of left pedal phalanges), Kol (MPC 100/2001; Turner, Nesbitt & Norell, 2009: left pedal phalanx III-2) and Albinykus (MPC 100/3004; Nesbitt et al., 2011: right pedal phalanx IV-3). A similarly narrow and less developed intercondylar groove is found on the distal end of the left pedal phalanx II-1 of the Linhenykus paratype specimen (IVPP V18190; Hone et al., 2013), but a wider and more developed groove is found in the holotype specimen (IVPP V17608; Xu et al., 2013b). This degree of variability implies that the element in question might be impossible to identify based on this characteristic alone. Alternatively, this morphological difference might be of taxonomic importance, although differentiating between this scenario and the former one is beyond the scope of this paper. A similarly developed narrow intercondylar groove appears to be present on the distal end of the right pedal phalanges II-1 and IV-1 of Albinykus (MPC 100/3004; Nesbitt et al., 2011), but which of these the element most closely resembles overall is unclear. There is a well-developed, anteriorly-marginated, triangular-shaped ligamental fossa on the lateral surface of the lateral condyle of this IVPP V20341 element, but this area is poorly preserved in the aforementioned phalanges of Albinykus. In lateral view, the ventral surface of the lateral condyle of this IVPP V20341 element is deeper and more steeply inclined than its dorsal surface. Neither right pedal phalanges II-1 or IV-1 of Albinykus have this feature, although a more weakly developed version is present in the former phalange. Therefore, the digital element in question is speculatively identified as a pedal phalanx II-1. This phalanx is potentially from the right foot because the larger condyle is the lateral rather than medial one, unlike the left pedal phalanx II-1 of Linhenykus (IVPP V17608; Xu et al., 2013b; IVPP V18190; Hone et al., 2013). However, the opposite could also be inferred since the reversed pattern is present in the left pedal phalanx II-1 of Mononykus (MPC 107/6; Perle et al., 1994). The shaft of the IVPP V20341 element is incomplete, but it does appear to be relatively straight. This feature would appear to rule it out as a metatarsal III, because the only taxon where the distal articular surface of MTIII has distinct condyles separated by a narrow intercondylar groove is Alnashetri (MPCA 477), but this has a shaft with an anteriorly convex curvature. Albertonykus (TMP 2001.45.52) and Linhenykus (IVPP V17608) have a similar shaft curvature, but the intercondylar groove is weakly developed in Albertonykus (TMP 2000.45.12; Longrich & Currie, 2009) and absent in Linhenykus (IVPP V17608, IVPP V18190; Xu et al., 2013b).

Potential right pedal digit II-2 phalanx

A potential right pedal digit II-2 phalanx is preserved (Fig. 5B). Its anterior articular surface has a simple concave morphology indicative of a more anteriorly-located phalanx, as in the right pedal phalanx II-1 of Albertonykus (TMP 2000.45.61; Longrich & Currie, 2009). However, the narrow width of the phalanx is seemingly at odds with this inference. The thicker and more robust medial side of the anterior articular surface rim suggests that it belongs to a right phalanx, as in the right pedal phalanx II-1 of Albertonykus (TMP 2000.45.61; Longrich & Currie, 2009). The partial dorsal surface of the phalangeal shaft rises up to the dorsal rim of the anterior articular surface more steeply than the lateral surface of the shaft rises up to the lateral rim of the anterior articular surface. As a right pedal phalanx II-1 has been suggested already (Fig. 5A), this element could be from the II-2 position instead.

Possible pedal phalanx from the second or third digit

A reasonably anteroposteriorly long but dorsoventrally low phalanx is preserved with a shallowly sinking ventral surface and a dorsal surface with a broad ridge that traverses it diagonally. This potentially identifies this element as a pedal phalanx from the second or third digit (Fig. 5C), although this element might be too small to fit this identification.

Potential left pedal phalanx III-2/3

The study of this element and the following one (left pedal phalanx III-3) suggests that the former is potentially a left pedal phalanx III-2. Based on how the possible pedal phalanx II-1 was assigned to the right side of the body, it follows that the larger of two concavities that are separated by a vertical ridge on the anterior articular surface of this element potentially corresponds to the lateral side of this surface (Fig. 5D). This suggests that this element belongs to the left side of the body. However, as mentioned, this characteristic varies between taxa (Linhenykus (IVPP V17608; Xu et al., 2013b; IVPP V18190; Hone et al., 2013) compared to Mononykus (MPC 107/6; Perle et al., 1994)) and seeming along a single digit as well (along left pedal digits III and IV of Mononykus (MPC 107/6; Perle et al., 1994)). The concave articular facets of the anterior articular surface do not extend across the entire dorsoventral height of the articular surface, but meet a flat facet approximately two-thirds down this surface. The dorsal portion of the vertical ridge extends slightly anteriorly to overhang the anterior articular surface. However, this is far less extensive than in phalanx IV-4 of Linhenykus (IVPP V17608; Xu et al., 2013b) where this happens for the dorsal and ventral portions of the ridge, dividing the entire dorsoventral height of the anterior articular surface. These aforementioned features indicate a more posteriorly positioned phalanx from perhaps the second or third positions. The shaft of the phalangeal element is only partially preserved, but it appears to have a rounded ventral surface. The lateral condyle of the posterior articular surface is missing, but the medial condyle is well-developed and is bound laterally by a well-formed intercondylar groove. This condyle has a strongly asymmetric lateral profile with a rounded dorsal surface and an elongated sloping ventral surface, as in many theropod pedal phalanges including those of Deinonychus (YPM 5205; Ostrom, 1969). This feature is present in a much less developed condition in phalanx IV-4 of Linhenykus (IVPP V17608; Xu et al., 2013b), but this phalanx has a stouter profile than IVPP V20341 (the latter phalanx is much longer anteroposteriorly than tall dorsoventrally compared to the former phalanx) (Xu et al., 2013b). A well-developed ligamental fossa occurs slightly below the mid-point of the condyle’s lateral surface. The relative slenderness of the preserved phalanx suggests that it belongs to the third rather than fourth digit. Thus, the phalanx concerned is potentially the second or third position of a left pedal digit III (?left III-2/3).

Potential left pedal phalanx III-3

A fragment of the anterior portion of a potential left pedal phalanx III-3 is preserved (Fig. 5E). This shares a similar asymmetrical anterior articular surface morphology as the possible left pedal III-2/3 phalanx, indicating that it shares a similar position along the digit and belongs to the same side of the body. However, the taller dorsoventral height of the articular surface and the more subtriangular outline of its dorsal edge, suggests that it is more anteriorly-located than the left pedal III-2/3 phalanx. This is also indicated by the prominently projecting ventrolateral corners of the left pedal phalanx’s anterior articular surface, instead of the rounded ventral margin of the anterior articular surface of the left pedal phalanx III-2/3. Thus, this element could correspond to a left pedal phalanx III-3 whereas the previous phalanx could be a left pedal phalanx III-2 instead.

Possible right pedal phalanx possibly from digit IV

An anteroposteriorly long phalangeal element with a broad ridge along the dorsal surface of its shaft and an expanded anterior corner on its right lateral side (Fig. 5F) is possibly from a right pedal digit IV. Its anterior articular surface is similar in form to the potential left III-2 and III-3 phalanges, which also have unequally-sized concavities. However, following the logic used and discussed above, the position of the larger concavity on the right lateral side of the element—even though this facet is partly damaged—indicates that this phalanx might be from the right side of the body. The anterodorsal portion of the phalanx is laterally pinched (subtriangular outline) whilst the posterior end of the phalanx is dorsoventrally depressed. These features could identify the phalanx as an element from digit IV, but this is speculative.

Potential pedal phalanx from digit IV

A potential pedal phalanx from digit IV is identified based on its seemingly short anteroposterior length, its apparently asymmetrical condyle in lateral view (as in the possible left pedal phalanx III-2 described above) and a shaft with a steeply lowering ventral surface (Fig. 5G). These features resemble those of the ?right pedal phalanx IV-4 of Linhenykus (IVPP V17608; Xu et al., 2013b), but the element in question is too poorly preserved for both its position along the digit and its side of the body to be suggested.

Discussion

IVPP V20341 is referable to Alvarezsauroidea because of the absence of cervical epipophyses (absent above the left lateral postzygapophysis of anterior cervicals B and C (Fig. 2A); character state 6.1 of Longrich & Currie (2009) is an alvarezsauroid synapomorphy, but the mid-cervicals of Linhenykus have weakly developed ridge-like epipophyses (Xu et al., 2013b)). The specimen is an alvarezsaurid alvarezsauroid owing to the presence of cervical centra bearing deep lateral depressions (Xu et al., 2013b) (centra of anterior cervicals A–C and the isolated anterior cervical (Fig. 2); character state 8.1 of Longrich & Currie (2009)). This placement is also supported by the presence of caudal procoely (Fig. 3; character state 21.1 of Longrich & Currie (2009)). Owing to the incomplete preservation of the pedal digits in IVPP V20341, it is unclear if pedal digit III is more slender than digits II or IV, so this alvarezsaurid synapomorphy (Xu et al., 2013b) cannot be confirmed in this specimen. IVPP V20341 is further identified as a parvicursorine alvarezsauroid based on the presence of anterior caudal vertebrae with anteriorly displaced transverse processes (Xu et al., 2013b) (Fig. 3; character state 22.1 of Longrich & Currie (2009)).

Currently, only one parvicursorine—Linhenykus monodactylus Xu et al. 2011—is known from the same locality and formation as IVPP V20341 (Bayan Mandahu, Inner Mongolia, China; the Upper Cretaceous—possibly Campanian—Wulansuhai Formation (Xu et al., 2010b; Eberth, 1993; Jerzykiewicz et al., 1993)). However, six other parvicursorines are known from more northerly localities within the Santonian to Maastrichtian-aged Upper Cretaceous rocks (Alifanov & Barsbold, 2009; Karhu & Rautian, 1996; Eberth et al., 2009; Gao & Norell, 2000; Jerzykiewicz & Russell, 1991; Lillegraven & McKenna, 1986) of the Mongolian Gobi Basin: Albinykus (Nesbitt et al., 2011), Ceratonykus (Alifanov & Barsbold, 2009), Mononykus (Perle et al., 1994), Parvicursor (Karhu & Rautian, 1996), Shuvuuia (Chiappe, Norell & Clark, 1998) and Kol (Turner, Nesbitt & Norell, 2009) (Table S1). Agnolin et al. (2012) argue that Kol has stronger oviraptorosaurian affinities than alvarezsaurid ones, but having not studied the specimen yet first-hand, we adopt the original identification here. IVPP V20341 does not have any known autapomorphies of any other parvicursorine and its unique features (possible procoelous cervicals and laterally wider and more semi-circular caudal neural canals) are insufficient to assign it to a new species because they can potentially be explained as anatomical variations along the vertebral column (particularly as the latter is poorly understood amongst parvicursorines). A better understanding of anatomical variation in Linhenykus in the future might led to IVPP V20341 being referred to this taxon, but current evidence does not permit such a referral.

Parvicursorines at Bayan Mandahu

The period of deposition represented at Bayan Mandahu is not known accurately, but some lithologies such as the structureless sandstones seem to have been rapidly deposited by sandstorm events, whilst others such as the carbonates (caliche) were probably deposited more slowly over thousands of years (Jerzykiewicz et al., 1993). This suggests that deposition probably happened over thousands of years at Bayan Mandahu, although this cannot be confirmed until radiometric dating can establish a chronology for the local depositional history. Given this estimate, the locations of IVPP V20341 and Linhenykus specimens (IVPP V17608, IVPP V18190) near the top and bottom of the local rock succession (Fig. 1) suggests that the deposition of both taxa was probably separated by a similar magnitude of time. This time interval perhaps makes it more likely that IVPP V20341 belongs to Linhenykus, but it is possible that there were two or more distinct genera in the locality that had separate and/or overlapping temporal ranges. IVPP V20341 and Linhenykus were preserved under broadly similar semi-arid conditions—the former is preserved in a red structureless sandstone layer whilst the latter is preserved in a more resistant nodule-rich red sandstone layer (Table S1). This indicates that IVPP V20341 and Linhenykus lived in a similar environment within or close to a dune field, according to Eberth’s (1993) depositional model for Bayan Mandahu (alluvial material washed off the nearby palaeo-Lang Shan mountain range was fringed by a dune environment). The persistence of environmental conditions potentially favours the longevity of an existing genus, but this likelihood cannot be used to justify IVPP V20341’s taxonomy. Thus, there are contextual arguments for the referral of IVPP V20341 to Linhenykus but these are seemingly weak ones. The study of the specimens themselves demonstrates that IVPP V20341 is distinguishable from Linhenykus monodactylus by a feature of the anterior caudal vertebrae: transverse processes originate from the dorsoanterior corner of the centra (anterior caudals A and B) rather than the posterior end of the prezygapophyses, as in Linhenykus (e.g., caudals 3, 5, 7 and 8 of IVPP V17608; Xu et al., 2013b). This is because the first caudal of Linhenykus is ventrally-keeled but neither caudals A or B are.

There are a number of suspected differences between the cervical and caudal vertebrae of IVPP V20341 and Linhenykus that have also been discussed. However, these must remain as such until the vertebral number of the relevant vertebrae can be clarified—particularly by further parvicursorine discoveries. Owing to the uncertainty in the identification of elements from the appendicular skeleton, their differences with Linhenykus are also excluded here.

IVPP V20341 compared to other Asian parvicursorines

∼350 km separates Bayan Mandahu and the closet Mongolian parvicursorine locality (Mononykus olecranus at Bayan Dzak (Andrews, 1932)). During the Upper Cretaceous the mountain ranges within the Gobi Basin (composed of Palaeozoic and Mesozoic rocks) were being subjected to extensional tectonism (Jerzykiewicz et al., 1993) that presumably promoted sediment deposition through the creation of accommodation space. This geological setting divided the Gobi Basin and created obstacles to faunal interaction, which probably promoted vicariance. The latter would help to explain why the Bayan Mandahu fauna seems to be distinct from Djadokhtan ones (Xu et al., 2015). If common Bayan Mandahu and Djadokhtan parvicursorines were found, this would suggest that at least some elements of the faunas are similar, which would advocate a complex scenario of selective isolation to explain the pattern of animals observed. Fortunately, for the skeletal elements that they share in common, IVPP V20341 lacks the autapomorphies of any Mongolian parvicursorine, so this material does not appear to contradict the distinctiveness of the Bayan Mandahu fauna (see Table 3). However, its status as a distinct taxon or specimen of Linhenykus remains unclear. Parvicursor and Ceratonykus are both known from the Upper Cretaceous (Lower Santonian, Alifanov & Barsbold, 2009; Middle Campanian, Karhu & Rautian, 1996) Barun Goyot Formation (Table S1) and appear to have shared their living environment. Niche partitioning by these taxa—if at all— probably relates to their body size differences as Parvicursor is smaller than Ceratonykus (75.6 mm long tibiotarsus in Parvicursor, PIN 4487/25 (Karhu & Rautian, 1996); 89 mm long right and left tibiotarsus in Ceratonykus, MPC 100/124 (Alifanov & Barsbold, 2009)). If IVPP V20341 is shown to be a valid taxon at a later date, the sharing of a relatively harsh semi-arid environment with Linhenykus might support niche partitioning too e.g., if IVPP V20341 has a more conventional parvicursorine hand morphology than Linhenykus (Xu et al., 2013b).

Table 3 Diagnoses of Asian parvicursorines.

None of the osteological features listed in the diagnoses of Asian parvicursorines are present in IVPP V20341. Features belonging to skeletal elements that are also preserved in IVPP V20341 are in bold font. See Table S1 for additional taxon information.

Taxon	Diagnosis	
Albinykus (Nesbitt et al., 2011)	Possesses a unique character state combination amongst alvarezsaurids (from Nesbitt et al. (2011)):	
	• Short metatarsal I with a rounded anterior tip (unknown in both Alvarezsaurus and Patagonykus).	
	• Well-pronounced and knob-like crest on fibula (attachment site for the M. iliofibularis) proportionally larger than other alvarezsaurids.	
	• Phalanx IV-4 longer than both phalanges IV-2 and IV-3.	
	∘ Equivocal in IVPP V20341 as phalanx IV-2 is missing.	
	• Deep groove present on the anterior face of the ascending process of the astragalus.	
	• A small flange on the lateral side of the distal end of metatarsal IV shared with Parvicursor,	
	Shuvuuia and Mononykus only.	
Ceratonykus (Alifanov & Barsbold, 2009)	From Alifanov & Barsbold (2009):	
	• Preorbital skull region long.	
	• Upper temporal fenestrae ovate, 0.4 as long as frontals.	
	• Length of one frontal almost four times greater than its width.	
	• Frontals narrowing rostrally in narrow wedge.	
	• Prefrontals adjoining each other medially.	
	• Basipterygoid processes two-thirds as high as quadrates.	
	• Labiooccipitally, dentaries forming deep and rostrally tapering depression.	
	• Mandibular fenestrae extensive.	
	• Centra of cervical and anterior caudal vertebrae narrow.	
	∘ No relative measure of narrowness is provided so this characteristic is difficult to confirm in IVPP V20341, especially when the latter lacks a suitable body proxy at present.	
	• Deltopectoral crest separated from humeral head by notch.	
	• Basal phalanx of major digit of manus extended, its flanks are moderately wide and the distal condyle is narrow and symmetrical.	
	• Postacetabular plate of ilia with relatively small longitudinal craniomedial crest.	
	• Femora strongly curved, nearly half as long as tibiotarsus.	
	• Fourth trochanter distinct.	
	• Cnemial crest of tibiae undeveloped.	
	• Ascending process of astragali high and wide.	
	• Tarsometatarsals 1.33 as long as femora.	
	• Second and fourth metatarsals tightly adjoining each over entire extent; their dorsal and palmar surfaces ridge-like, deep grooves formed between these bones. Deep notch formed anterodorsally between these metatarsals.	
	• Distally, second metatarsals shorter than fourth.	
	• Tarsometatarsals 3.5 times as long as third metatarsals.	
	• Basal phalanx of fourth digit of hind feet only slightly shorter than basal phalanx of second digit.	
	∘ Basal phalanges of the second and fourth digits are missing in IVPP V20341.	
Linhenykus (Xu et al., 2010b)	Distinguished from other parvicursorines by (from Xu et al. (2013b)):	
	• Transversely compressed metacarpal III without a distal articular surface.	
	• Longitudinal ventral furrow along the entire length of each cervical centrum.	
	∘ Rounded ventral surface in anterior cervicals of IVPP V20341.	
	• Diapophyseal ridges on each middle cervical vertebra extend to the posterodorsal rim of the centrum.	
	∘ Extend to the posteroventral rim in the anterior cervicals of IVPP V20341.	
	• Extremely weak, ridge-like epipophyses on the postzygapophyses of the middle cervical vertebrae.	
	∘ Epipophyses are absent in IVPP V20341 but the cervicals represent anterior ones only.	
	• Large pneumatic foramina in the mid-dorsal vertebrae.	
	• Anteriormost caudal vertebrae whose centra are amphiplatyan and whose neural spines are located completely posterior to the neural pedicles.	
	∘ All preserved caudals in IVPP V20341 are procoelous and the anteriormost caudal (caudal A) has a partial neural spine whose anterior margin appears to lie above the neural pedicle.	
Mononykus (Perle et al., 1993)	From Chiappe, Norell & Clark (2002):	
	• Cervical centra strongly compressed laterally, lacking pneumatic foramina.	
	∘ Pneumatic foramina also lacking in IVPP V20341 but its cervical centra are less strongly compressed mediolaterally.	
	• Cranialmost thoracic vertebrae strongly compressed.	
	• Fused ilium and ischium.	
	• Pillar-like deltopectoral crest of humerus.	
	• Supracetabular crest developed only in the cranial portion of acetabulum.	
	• Subtriangular cross-section of pubis.	
	• Two cnemial crest in tibiotarsus.	
	• Medial indentation of ascending process with deeply excavated base.	
	• Ascending process arises from medial margin of astragalar condyle instead of from lateral margin.	
Parvicursor (Karhu & Rautian, 1996)	From Chiappe, Norell & Clark (2002):	
	• Similar to Mononykus but smaller.	
	• Opisthocoelous caudal thoracic vertebrae.	
	• No bi-convex thoracic vertebra.	
	• Convex cranial margin of synsacrum.	
Shuvuuia (Chiappe, Norell & Clark, 1998)	Autapomorphies from Suzuki et al. (2002):	
	• An articulation between the quadrate and postorbital.	
	• Elongated basipterygoid processes.	
	• Hypertrophied prefrontal/ectethmoid.	
	• The presence of a sharp ridge on the medial margin of the distal tibiotarsus (Chiappe, Norell & Clark, 1998).	

IVPP V20341 compared to other alvarezsauroids

IVPP V20341 is seemingly distinct amongst alvarezsauroids because of possible cervical procoely and the presence of caudal neural canals (caudals A–C) that are laterally wider and more semi-circular compared to the laterally narrower and more oval-shaped ones of Linhenykus (caudals 2, 7 and 13) and of the supposed first caudals of Patagonykus (MCF-PVPH 37; Novas, 1997) and Parvicursor (Karhu & Rautian, 1996). However, these potential autapomorphies have caveats that need to be considered. Cervical procoely is unknown in alvarezsauroids, but only one complete neck specimen is known and this belongs to the basalmost taxon Haplocheirus sollers (Choiniere et al., 2010). If the eleven cervical vertebrae of the latter is similar amongst all alvarezsauroids—an assumption that is speculative based on current fossil evidence—then the 8 and 9 cervical vertebrae preserved in Mononykus (MPC 107/6; Perle et al., 1994) and Shuvuuia (MPC 100/975; Chiappe, Norell & Clark, 2002) respectively may actual represent near complete series. These three neck specimens provide an indication of the neck morphology in basal alvarezsauroids and derived parvicursorines—the latter of which IVPP V20341 should most closely resemble—so the absence of procoely in all three specimens is significant. However, Haplocheirus lacks the strong opisthocoelous condition of Mononykus (MPC 107/6; Perle et al., 1994) and Shuvuuia (MPC 100/975; Chiappe, Norell & Clark, 2002), which shows that there is significant variation in articular surface morphology within the clade, although it is currently impossible to say if such variation might include multiple taxa with a procoelous condition. Alvarezsauroid tail morphology is better understood than neck morphology because the former is known from more fossil material belonging to a broader range of taxa. The most complete caudal series are found in Haplocheirus (IVPP V15988, 15? caudals (Choiniere et al., 2010)), Alvarezsaurus (MUCPv 54, 13 caudals (Chiappe, Norell & Clark, 2002; Bonaparte, 1991)), Linhenykus (IVPP 17608, 13 caudals (Xu et al., 2013b)) and Shuvuuia (MPC 100/975, 19 caudals (Chiappe, Norell & Clark, 2002); MPC 100/120; 22? caudals (Suzuki et al., 2002)). The most complete alvarezsauroid tail is represented by specimen MPC 100/120 of Shuvuuia, which preserves direct evidence of approximately 22 caudals (Suzuki et al., 2002). However, the gaps in the caudal series suggest a caudal count upwards of 35 caudals (Suzuki et al., 2002). The relatively large semi-circular caudal neural canals of IVPP V20341 are absent in Linhenykus, Patagonykus and Parvicursor—the only taxa that had specimens where the shape of the caudal neural canal could be determined from firsthand study or from the literature. This represents a small sample size and given that the neural canal varies in size and shape along the vertebral column of theropods (and other vertebrates), this potential autapomorphy cannot be supported unequivocally. In the cases of both of these tentative autapomorphies of IVPP V20341, future fossil specimens are needed to test their validity.

Potentially informative features for alvarezsauroid phylogeny

New parvicursorine anatomical information obtained through the study of IVPP V20341 warrants editing of five Longrich & Currie (2009) phylogenetic characters (characters 3, 7–9 and 21) as well as the erection of a new character.

The possibly procoelous cervicals of IVPP V20341 are currently unique amongst alvarezsauroids which requires character 3 of Longrich & Currie (2009) to be edited. Cervical procoely is probably a derived alvarezsauroid condition since the majority of alvarezsauroids have opisthocoelous cervicals and the basal condition seems to be amphicoelous, amphiplatyan or platycoelous (amphi- to platycoelous in Haplocheirus (IVPP V15988; Choiniere et al., 2010) and amphiplatyan in Alvarezsaurus (MUCPv 54; Chiappe, Norell & Clark, 2002; Bonaparte, 1991)). However, given the unknown combinations of these vertebral types in alvarezsauroid necks and their changes through time, this character is not ordered here:

Cervical centra: amphicoelous, amphiplatyan or platycoelous (0), opisthocoelous (1), procoelous (2), amphicoelous, amphiplatyan or platycoelous AND opisthocoelous or procoelous (3), amphicoelous, amphiplatyan or platycoelous AND opisthocoelous AND procoelous (4) (after Perle et al., 1994).

At a qualitative level, we observed noticeable changes in the ventral surface width of parvicursorine cervicals along their series. This suggests that further quantitative study is needed to maximize the phylogenetic utility of this feature and build upon the ordered character 7 of Longrich & Currie (2009).

As mentioned, the lateral sides of the cervicals of IVPP V20341 are less depressed than those of Linhenykus (IVPP 17608; Xu et al., 2013b), and this depression is limited to the ventrolateral portion of the centra. However, the position of these cervicals may not be analogous. To accommodate this difference and the expected variability in the degree of lateral surface depression along the neck of parvicursorines—pending more in-depth quantitative studies—character 8 of Longrich & Currie (2009) was edited:

Lateral surfaces of cervical centra: convex or flat (0), strongly to mildly depressed across part of or the entire surface (1), convex or flat AND strongly to mildly depressed across part of or the entire surface (2).

Comparisons made between the preserved cervical and caudal vertebrae of IVPP V20341 and all other parvicursorines has highlighted variation in the ventral surface along each series, including the relative development of furrows (partly or fully) and keels (absent, small or large in caudals) as well as the distribution of rounded and/or flat smooth ventral surfaces. To reflect these observations, character 9 of Longrich & Currie (2009) has been reworded:

Ventral surfaces of cervical centra: smooth and flat and/or smooth and rounded (0), longitudinal furrow partly or fully spanning the length of the centrum (1), both conditions are present (2) (after Novas, 1996).

Character state 2 is added because current evidence cannot rule out the possibility that states 0 and 1 are present in the same cervical series. However, the character remains unordered in the absence of evidence regarding how this trait evolved across Alvarezsauroidea.

In consideration of the amphiplatyan first caudal of Linhenykus (IVPP 17608; Xu et al., 2013b) and the amphicoelous (biconcave) proximal (4?) caudal of Achillesaurus (MACN-PV-RN 1116; Martinelli & Vera, 2007), character 21 of Longrich & Currie (2009) is expanded to:

Caudal vertebrae: amphiplatyan or amphicoelous (0), or procoelous (1) (after Novas, 1996).

To utilise the potential of the caudal ventral keel towards reconstructing alvarezsauroid phylogeny whilst considering their poorly known extent along the tail, a new character limited to the first caudal is proposed:

Ventral surface of the first caudal vertebrae: not transversely narrow (0), ‘pseudo-keel’ present—the ventral surface is transversely narrow and slightly flat (1), sharp keel present (2).

Bayan Mandahu as a distinct fauna within the Upper Cretaceous Gobi Basin

The Wulansuhai Formation rocks of Bayan Mandahu, Inner Mongolia comprise of lithologies that are similar to the Djadokhta Formation rocks of Bayan Dzak, Mongolia (Eberth, 1993; Jerzykiewicz et al., 1993). These lithologies indicate that both formations were deposited mostly under semi-arid conditions as alluvial and aeolian sediments, but the presence of some mudrocks shows that some deposition occurred under wetter climatic conditions (Eberth, 1993; Jerzykiewicz et al., 1993). Many Bayan Mandahu fossils have been referred to taxa known from the Djadokhta Formation (Jerzykiewicz et al., 1993; Currie & Peng, 1994; Dong & Currie, 1996), which both share a vertebrate fauna of dinosaurs, lizards, turtles, mammals and birds. The Wulansuhai Formation was assigned a Campanian age based on its lithological and faunal similarities with the Campanian-aged Djadokhta Formation (Jerzykiewicz et al., 1993), which itself was dated based on faunal and magnetostratigraphic data (see Xu et al. (2015) and references therein). The absolute age of the Wulansuhai Formation is still wanting so the stratigraphic correlation of these formations remains equivocal. However, an increasing body of evidence suggests that the two faunas represented in both formations are actually distinct (Makovicky, 2008): several previous referrals of Bayan Mandahu specimens to Djadokhta taxa have been rejected (Xu et al., 2012; Longrich, Currie & Dong, 2010) whilst several taxa unique to Bayan Mandahu have been described (Xu et al., 2010b; Godefroit et al., 2008; Longrich, Currie & Dong, 2010; Xu et al., 2010a; Xu et al., 2013a; Xu et al., 2013b). Unfortunately, the taxonomic status of IVPP V20341 does not contribute strong support towards the hypothesis that Bayan Mandahu is faunally distinct from the Djadokhta Formation, but it does not appear to contradict it.

Conclusions

A new parvicursorine alvarezsauroid theropod specimen IVPP V20341 from the Campanian-aged rocks of Bayan Mandahu, Inner Mongolia, China is described. This specimen has a different origination point for the anterior caudal transverse processes compared to the only other parvicursorine from this locality—Linhenykus. However, there are a larger number of tentative differences that require information from future finds to confirm—particularly with regards to anatomical variation along the parvicursorine spine. IVPP V20341 lacks any of the known autapomorphies of other Asian parvicursorines, but this is partly because many relevant elements are missing from the specimen. IVPP V20341 is seemingly unique amongst alvarezsauroids because of possible cervical procoely and the presence of relatively larger semi-circular caudal neural canals. However, these features can be plausibly explained as anatomical variations of the parvicursorine cervical and caudal series: articular face geometry actually varies considerably along the alvarezsauroid dorsal and caudal series, whilst neural canal anatomy varies in both size and shape along the vertebral column of theropods (and other vertebrates). Thus, erring on the side of caution, IVPP V20341 is not identified as a new taxon here, although more complete knowledge of the parvicursorine vertebral column arising from future discoveries may warrant a taxonomic revision. As a parvicursorine specimen without any autapomorphies, IVPP V20341 does not contradict the hypothesis that the Bayan Mandahu fauna is unique compared to other localities within the Upper Cretaceous Gobi Basin. Thus, despite the description of this specimen there are still seven parvicursorine species in the latter basin.

Supplemental Information

Table S1 Alvarezsauroid taxon data

Taxon data for known alvarezsauroid theropods.

Click here for additional data file.

Table S2 Alvarezsauroid body size estimates

Estimates of alvarezsauroid theropod body size and their associated measurements.

Click here for additional data file.

Figure S1 Unidentified IVPP V20341 locality bones

Unidentifiable bone fragments from the IVPP V20341 locality, including a probably partial centrum and potential tarsal (mammal?) or carpometacarpal bones.

Click here for additional data file.

The authors wish to thank all of the members of the 2013 Inner Mongolia Research Project (IMRP) team (including Zhao Qi and Corwin Sullivan) and Ding Xiaoqin for preparing the specimen.

Additional Information and Declarations

Competing Interests

Author Contributions

Field Study Permissions

The authors declare there are no competing interests.

Michael Pittman conceived and designed the experiments, performed the experiments, analyzed the data, contributed reagents/materials/analysis tools, wrote the paper, prepared figures and/or tables, reviewed drafts of the paper.

Xing Xu conceived and designed the experiments, performed the experiments, analyzed the data, contributed reagents/materials/analysis tools, wrote the paper, reviewed drafts of the paper.

Josef B. Stiegler conceived and designed the experiments, analyzed the data, reviewed drafts of the paper.

The following information was supplied relating to field study approvals (i.e., approving body and any reference numbers):

A fossil excavation permit was obtained from the Department of Land and Resources, Linhe, Inner Mongolia, China. This permit allowed the authors and other Inner Mongolia Research Project team members to extract and study material from our field site.

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
