# Peer review of "The taxonomy of a new parvicursorine alvarezsauroid specimen IVPP V20341 (Dinosauria: Theropoda) from the Upper Cretaceous Wulansuhai Formation of Bayan Mandahu, Inner Mongolia, China"

_PeerJ, doi:10.7717/peerj.986_

## Round 0.1 · original submission · Minor Revisions

This paper is an important description for a member of a rare taxonomic group, and is undoubtedly a worthy contribution to the literature. I applaud your caution in dealing with the taxonomic assignment for the specimen, rather than erecting a new (and potentially invalid) species. However, I do agree with the anonymous reviewer that the way this conclusion is presented seems somewhat contradictory; please revise the manuscript for clarity along these lines. Both reviewers note, and I agree, that the figures could be improved, particularly in the way that the bone outlines are cropped. If the specimen is still in matrix, it might be visually clearer to simply leave the matrix in the image. Otherwise, please re-edit the images with a closer eye to visual clarity and faithfulness to the original element. The reviewers include a number of other, more minor, suggestions that should also be addressed in revision.

- I recommend splitting the measurement tables--vertebral measurements in one, appendicular elements in the other.

·

Basic reporting

17-12-2014
Dear Editor,
Here I am sending the reviewed manuscript entitled as “The taxonomy of a new parvicursorine alvarezsauroid specimen IVPP V20341 (Dinosauria: Theropoda) from the Upper Cretaceous Wulansuhai Formation of Bayan Mandahu, Inner Mongolia, China” authors M. Pittman, X. Xu and J. Stiegler. The article contains valuable information about parvicursorine material, and consists an important addition to the knowledge of the clade. Although I am not a native English speaker I think that the manuscript needs an important improvement on its language. In addition, some modifications to the figures (specially the cropping of the images) should be added. I have attached some minor comments in the PDF that I am attaching. I think that the manuscript must be published after the revision of minor details,
All the best,

Federico Agnolin
Laboratorio de Anatomía Comparada y Evolución de los Vertebrados

Experimental design

No comments

Validity of the findings

No comments

Reviewer 2 ·

Basic reporting

Generally well-written paper, publishable with a few minor, but critical edits.

The figure quality must be improved. The elements have coarse edges, floating pixels, and it appears as if some of the elements have been cut into. As shown, it is difficult to identify the real margins of the elements. Also, why is the ventral view of the four articulated cervicals truncated in Figure 2? Please illustrate the rounded, not grooved, ventral margin, because this is a significant part of the description.

Experimental design

The authors present a detailed and comprehensive differential description of this new specimen. I have no technical issues with the description; however I would present one suggestion to the authors that would benefit the reader. Clarity of the description would improve if the authors were to present their tentative identification of each element upfront and then describe the element in light of this identification. As written, the identification of the elements seems to be up-in-the air until the end of the description, which is confusing. The authors take the reader step-by-step through the identification process, rather than simply describing the element as they have identified it, even if that identification is tentative.

Validity of the findings

The two conclusions presented in this paper are contradictory. The authors present possible autapomorphies of IVPP V20341 and list numerous (13) differential features, but then go on to conclude that this specimen cannot reasonably be considered a unique taxon, presumably because the vast majority of differential features reflect the morphology of non-comparable cervical vertebrae. Either the conclusion that this taxon can be distinguished by cervical proceoly, a larger neural canal, and 13 other axial features is warranted by the data at hand and should stand as a main conclusion, or the authors should stand by their identification of the cervicals of IVPP V20341 as anterior, and therefore not directly comparable to the middle and posterior cervicals of Linhenykus, which negates their two potential cervical autapomorphies and 9 of the 13 differentiating features of the cervical series. To present these features as differential, and then go on to say they cannot be considered reliable because the vertebrae are from a different part of the cervical series is misrepresentative and unnecessarily confusing to the reader. To avoid misrepresentation, please decide which one of these conclusions best represents the data at hand and re-work the paper to reflect this decision. My interpretation would be to remove the features of the non-equivalent cervical vertebrae from the differentia.

Additional comments

specific comments:

Sys paleo, material section - Replace broken with fragmented, fragmentary, partial etc… as appropriate

- line 79…. What is a more ontogenetically mature individual? Reword. Perhaps an individual at or near skeletal maturity?

-replace “like” throughout, replace with “as in” “such as” etc…

Lines 459, 461 - pedal “phalanx”, not “phalange” (check throughout)

Line 561-571 – when using italics for entire sentences that include taxon names, taxon names should not be in italics.

Line 586 - if you aren’t going to describe differences in the appendicular skeleton, why have the heading? Please remove.

Line 704 – comma before “which” (check throughout)

-authors use both anterior/posterior and proximal/distal to describe the same orientation in the axial column. Better to avoid using proximal/distal as general orientation for the axial column terms unless the context is absolutely clear…

Line 540 - authors state that 16 anatomical characteristics differ between Linhenykus and IVPP20341, yet list only 13. And the first characteristic (line 544) is written as if it is a shared feature.

The authors tentatively identify the preserved cervical vertebrae as anterior (an identification they make a good argument for and that I support), yet fail to use this identification throughout the paper in a meaningful way.

The discussion of the differences in the cervical series between Linhenykus and IVPP20341 are meaningless if we accept that these vertebrae are from different regions of the cervical series. Most of these features are entirely consistent with anterior vs posterior cervicals of parvicursorines. For example, carotid processes would be expected in posterior cervicals and expected to be absent in anterior cervicals. It would seem that rather than providing useful differentiation between these specimens, this section provides a morphological argument for why these cervicals are interpreted as anterior, rather than mid-or posterior.

If the cervicals are anterior and therefore not comparable to the mid- and posterior cervicals of Linhenykus, than the main conclusion (line 719) that there are 13 anatomical differences (including 9 cervical differences) cannot be stated.

Table 2
Albinykus: (bold) phalanx IV-4 longer than both phalanges: you wrote “not equivocal” here, do you mean “equivocal?”
Ceratonkyus: (bold) basal phalanx of fourth digit of hind feet: if these elements are missing in IVPP V20341, why is this feature in bold? In the caption you state that bold features, reflect elements that are preserved with IVPP V20341.
Linhenkyus: again, I don’t see the point in listing differences between vertebrae of significantly different positions in the cervical series.

---

## Round 0.2 · Minor Revisions

Thank you for your close attention to the comments from the reviewers, particularly the work in improving the figures. I have looked over the manuscript, and it is nearly ready to go, but I noted one inconsistency in the abstract that should be addressed:

"IVPP V20341 is distinguishable from Linhenykus - the sole parvicursorine at Bayan Mandahu - by the location of the origination points of the anterior caudal transverse processes: in IVPP V20341 this is the anterodorsal corner of the centra whereas in Linhenykus it is the posterior end of the prezygapophyses. However, a number of tentative differences between IVPP V20341 and Linhenykus are also identified, but cannot be confirmed until further details of anatomical variation along the neck and tail are revealed by future finds. "

The use of "However" in the second sentence seems a little odd...would the following rewording be better?

IVPP V20341 is distinguishable from Linhenykus - the sole parvicursorine at Bayan Mandahu - by the location of the origination points of the anterior caudal transverse processes; in IVPP V20341 this is the anterodorsal corner of the centra, whereas in Linhenykus it is the posterior end of the prezygapophyses. A number of additional tentative differences between IVPP V20341 and Linhenykus are also identified, but cannot be confirmed until further details of anatomical variation along the neck and tail are revealed by future finds.

---

## Round 0.3 · accepted · Accept

Thank you for your quick attention to the abstract. In my view, the manuscript is ready to move forward.

---

## Author Rebuttal · Round 0.3

Dr. Michael Pittman,
Research Assistant Professor,
Vertebrate Palaeontology Laboratory,
Department of Earth Sciences,
The University of Hong Kong,
Pokfulam, Hong Kong.

mpittman@hku.hk

8th May 2015

Attn: Andrew Farke, Academic Editor for PeerJ

**Corrections to PeerJ submission 'The taxonomy of a new parvicursorine alvarezsauroid specimen IVPP V20341 (Dinosauria: Theropoda) from the Upper Cretaceous Wulansuhai Formation of Bayan Mandahu, Inner Mongolia, China'**

Dear Dr. Farke,

Thank you for your suggested edits to our manuscript's abstract; these edits have now been implemented.

Yours faithfully,

Dr. Michael Pittman (on behalf of Prof. Xing Xu and Mr. Josef B. Stiegler)